# Engineering eukaryote-like regulatory circuits to expand artificial control mechanisms for metabolic engineering in *Saccharomyces cerevisiae*

Bingyin Peng [1,2,3,4 ✉], Naga Chandra Bandari [1], Zeyu Lu [1], Christopher B. Howard[1], Colin Scott [2,5], Matt Trau [1,6], Geoff Dumsday[7] & Claudia E. Vickers [2,3,4,8 ✉]

Temporal control of heterologous pathway expression is critical to achieve optimal efficiency in microbial metabolic engineering. The broadly-used *GAL* promoter system for engineered yeast (*Saccharomyces cerevisiae*) suffers from several drawbacks; specifically, unintended induction during laboratory development, and unintended repression in industrial production applications, which decreases overall production capacity. Eukaryotic synthetic circuits have not been well examined to address these problems. Here, we explore a modularised engineering method to deploy new genetic circuits applicable for expanding the control of *GAL* promoter-driven heterologous pathways in *S. cerevisiae*. *Trans*- and *cis*- modules, including eukaryotic *trans*-activating-and-repressing mechanisms, were characterised to provide new and better tools for circuit design. A eukaryote-like tetracycline-mediated circuit that delivers stringent repression was engineered to minimise metabolic burden during strain development and maintenance. This was combined with a novel 37 °C induction circuit to relief glucose-mediated repression on the *GAL* promoter during the bioprocess. This delivered a 44% increase in production of the terpenoid nerolidol, to 2.54 g L$^{-1}$ in flask cultivation. These negative/positive transcriptional regulatory circuits expand global strategies of metabolic control to facilitate laboratory maintenance and for industry applications.

[1] Australian Institute for Bioengineering and Nanotechnology (AIBN), The University of Queensland, Brisbane, QLD 4072, Australia. [2] CSIRO Future Science Platform in Synthetic Biology, Commonwealth Scientific and Industrial Research Organisation (CSIRO), Black Mountain, ACT 2601, Australia. [3] ARC Centre of Excellence in Synthetic Biology, Queensland University of Technology, Brisbane, QLD 4000, Australia. [4] Centre for Agriculture and the Bioeconomy, School of Biological and Environmental Science, Queensland University of Technology, Brisbane, QLD 4000, Australia. [5] Biocatalysis and Synthetic Biology Team, CSIRO Land & Water, Black Mountain Science and Innovation Park, Canberra, ACT 2601, Australia. [6] School of Chemistry and Molecular Biosciences (SCMB), The University of Queensland, Brisbane, QLD 4072, Australia. [7] CSIRO Manufacturing, Clayton, VIC 3169, Australia. [8] Griffith Institute for Drug Discovery, Griffith University, Brisbane, QLD 4111, Australia. ✉email: bingyin.peng@qut.edu.au; claudia.vickers@csiro.au

Metabolic burden from expression of heterologous pathways may dramatically inhibit normal cell proliferation in microbial cell factories[1–3] and cause strain instability[1–5]. This delays laboratory strain development via the design-build-test-learn cycle and impedes industrial processes. It is critical to have robust control mechanisms that provide tight repression when optimal growth is required as well as boosted induction when maximal production is needed[6]. Importantly, these tools should not increase the cost for industrial processes.

The yeast *Saccharomyces cerevisiae* is a model Eukaryotic organism for understanding biological principles and a primary chassis organism used in manufacturing a variety of bioproducts[7,8]. The endogenous yeast galactose-inducible (*GAL*) expression system has been engineered to respond to glucose starvation for automatic induction of heterologous pathway[9,10]. In the presence of glucose, the glucose-dependent repressor Mig1p represses expression of alternative carbon source catabolic genes (including the *GAL* genes for galactose utilisation). When the *GAL* repressor gene *GAL80* is disrupted, *GAL* promoters are automatically induced upon glucose depletion in batch cultivation[2,9,11]. The low expression from *GAL* promoters in exponential growth phase on glucose prevents metabolic burden from causing growth inhibition. This engineered *gal80Δ GAL* induction circuit is now broadly used in metabolic engineering, as exemplified by high level production of terpenoids[12] and flavanones[13,14]. However, the system has limitations: (i) unintended auto-induction during routine strain maintenance and development is problematic when induced pathways result in metabolic burden or cellular toxicity; (ii) in prolonged pulse-feeding high-cell-density glucose processes, Mig1-mediated repression on *GAL* promoters may disrupt expression patterns; (iii) an increase in promoter strength is highly desirable to improve expression of bottleneck enzymes. The current work aims to address these limitations by expanding control range of *GAL* promoters for better application in metabolic engineering.

One of the applications of synthetic biology is in development of artificial regulatory circuits for application in metabolic engineering[15–17]. Many synthetic transcriptional circuits have been reported for *S. cerevisiae*[6,15,18–20]. They include VP16 *trans*-activating sequence-mediated activation circuits[19–21] and prokaryote-like bacterial repressor-mediated circuits[6,15]. A variety sensing mechanisms have been used to trigger the OFF/ON states of these circuits, i.e., small molecules[6,15,18–20,22], light[23–25], and cold -shock[26]. However, each circuit has specific down sides, which may make them non-ideal in application. For example, adding expansive small molecules is not economical for large-scale cultivation. In addition, cold-shock induction may not be possible in tropical regions, including Mackay, Queensland, one of Australia's major cane sugar feedstock processing regions. Current prokaryote-like circuits require high-level expression of bacterial repressors and intensive optimisation of promoters to achieve ideal on/off response ratio[6,15,27], which do not reach to the efficiency of natural eukaryotic regulatory mechanisms[28].

The complexity and relative unpredictability of biological systems still hinders rational design of genetic circuitry[6,15,29–34]. Consequently, significant work has been devoted to development of robust, predictable genetic componentry in yeast[6,35,36]. Comprehensive characterisation of individual modular components in different growth states is required to improve design parameterisation for these circuits. In this study, we employed a modularised engineering strategy for synthesis of synthetic regulatory circuits to expand control mechanisms on *GAL* promoters for metabolic engineering applications. We investigated eukaryote-like mechanisms to deploy *trans*-activation modules and *trans*-repression modules. We integrated new control mechanisms to render non-natural regulatory properties to the *GAL* expression system, including using tetracycline to trigger repression on *GAL* promoters and elevated temperature to trigger de-repression of *GAL* promoters. We also investigated possibility of applying artificial *trans*-activators to enhance *GAL* promoter strength. The expanded *GAL* system was then validated for repression or upregulation of heterologous sesquiterpene production in yeast.

## Results

**Native transcription factor promoters express at moderate to low levels.** A strong promoter, commonly used in overexpression of metabolic enzymes or primary cellular constitutive proteins, is often used to control the expression of artificial transcription factors (TFs) in synthetic circuits[15,35,37]. However, in natural eukaryotic systems, TFs are generally not expressed at high levels[38]. We wanted to explore whether we could achieve a better outcome by exploiting native TF promoters to mimic natural expression systems. We therefore characterised fourteen TF and regulatory protein promoters.

Yeast strains for promoter characterisation were obtained by transformation of an enhanced yeast green fluorescent protein (yEGFP) expression cassette under the control of promoter of interest *via* single-copy genome integration (Fig. 1a). Fluorescence was analysed in early exponential phase (when glucose was the carbon source) and in post-exponential phase (when ethanol was the carbon source).

We tested three base constructs using the *TEF1* promoter to identify the best one to use for promoter comparison (Fig. 1a). In our previous work[39], we used a construct with a modified translation initiation region including a *Bam*HI restriction enzyme site and a *URA3* terminator (which also acts as a homologous recombination site to introduce the construct onto the genome). Insertion of a *PGK1* terminator improved yEGFP fluorescence by >5-fold (Fig. 1b). removal of the *Bam*HI site and reinstatement of the native translation sequence context (Fig. 1a, Construct 3) further improved yEGFP fluorescence by ~50%. This construct design was used to test regulatory protein promoters.

Compared to the *TEF1* promoter (Fig. 1b, construct 3), the 14 tested endogenous regulatory protein promoters were weaker by 1–2 orders of magnitude (Fig. 1c). The promoters of the *GCR1* and *GCR2* genes, encoding transcriptional activators for glycolytic and ribosomal genes, were >39-fold weaker than the *TEF1* promoter, and the promoter of the *MIG1* gene, encoding an important repressor in the glucose repression signalling pathway, was further weaker. The promoters showing a similar or weaker strength include the promoters of *GLN3* (encoding a transcriptional activator in nitrogen catabolite repression system), *TOR1* (encoding a primary protein kinase in Target Of Rapamycin regulatory network), *SNF1* (encoding a primary protein kinase in glucose repression regulatory networks), *NRG1* (encoding a repressor mediating glucose repression), *ROX1* (encoding a repressor in oxygen regulation), *HAP4* (encoding a global regulator of respiration), and *UPC2* (encoding a ergosterol-sensing activator). This shows that many primary components in natural regulatory networks are expressed at the levels much lower than the levels achievable from strong constitutive promoters.

Among the tested promoters, the promoters of *YPK2* (encoding a protein kinase required for cell growth), *ADR1* (encoding a transcriptional activator in glucose repression), and *HAC1* (encoding a transcriptional activator regulating the unfolded protein response in endoplasmic reticulum) showed a moderate strength, around 10-fold lower than the *TEF1* promoter. The promoter of *DAL80* (encoding a negative

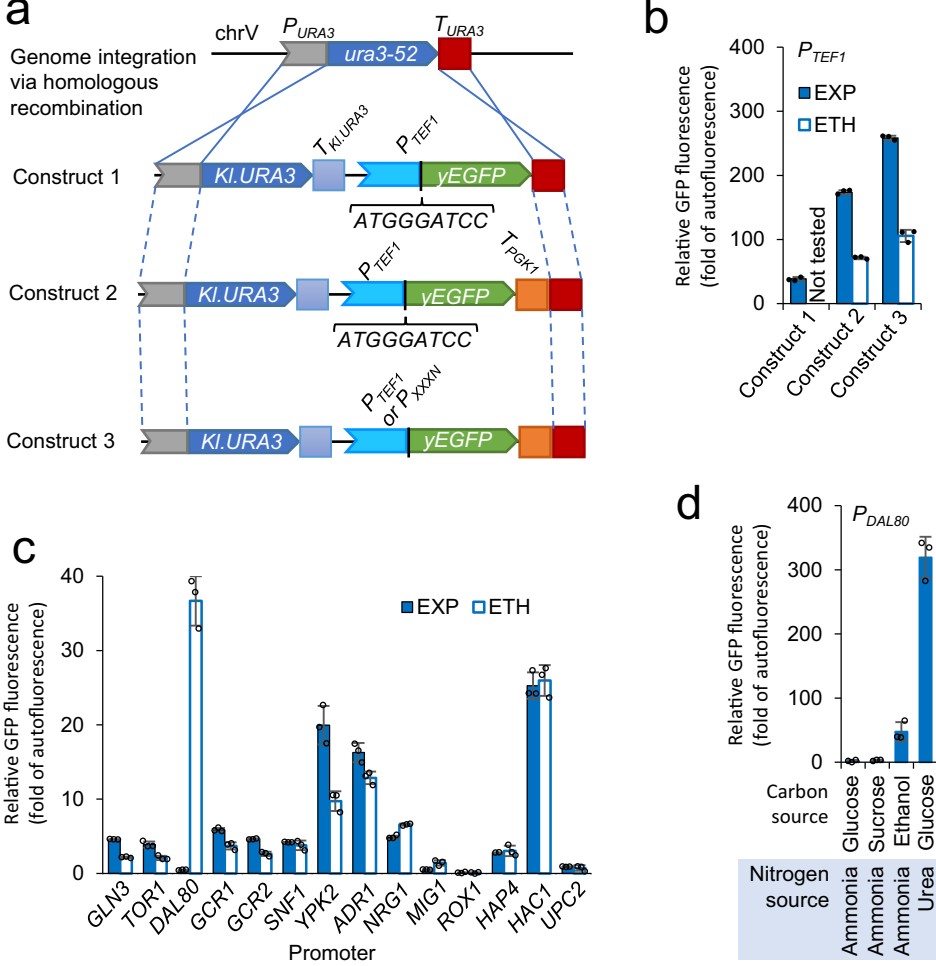

**Fig. 1 Characterisation of the promoters of endogenous regulatory protein genes. a** Schematic of the yEGFP reporter system used to examine the promoter strength. **b** The yEGFP expression levels from the *TEF1* promoter using the reporter system in **a**. **c** The yEGFP expression levels from the promoters of regulatory genes using construct 3 in **a**. **d** The yEGFP expression levels from the *DAL80* promoter under different carbon sources and nitrogen sources. **b**, **c** The yEGFP expression levels were characterised in the cells grown to the early exponential growth phase (EXP) or the ethanol-growth phase (ETH) in MES-buffered YNB media using 20 g L$^{-1}$ glucose as the carbon source in 96-well plates. **d** the yEGFP expression levels were characterised in the cells grown to the exponential phase in YNB media with varied combination of 20 g L$^{-1}$ glucose, 20 g L$^{-1}$ sucrose, or 2 % (v/v) ethanol as the carbon source and 5 g L$^{-1}$ ammonia sulphate or 0.46 g L$^{-1}$ urea as the nitrogen source. GFP fluorescence is expressed as percentage of exponential-phase auto-fluorescence of the reference strain. Data for Construct 1 in **b** are extracted from previous study[39]. Mean values ± standard deviations are shown ($N = 3$ independent biological replicates). Source data in **b**–**d** are provided in Supplementary Data 1.

repressor responsive to nitrogen levels) showed induced activities on ethanol (Fig. 1c, d), but not on sucrose (Fig. 1d). The *DAL80* promoter could be further induced to a level higher the *TEF1* promoter, when yeast was grown on urea, a poor nitrogen source (Fig. 1d). Due to the *HAC1* showing a consistent expression output on the exponential growth phase and the ethanol-growth phase and its moderate activity, we chose it to control artificial TFs for the following modular characterisation.

**Trans-activating modules have variable effects on transcriptional activity from different hybrid promotes**. The *trans*-activating domain (TAD) binds to transcriptional co-regulator proteins to refine and determine the magnitude of the response. Ottoz and colleagues[19] used a chimeric construct approach to characterise four TADs in yeast: VP16 (the alpha-gene-transactivating factor (α-TIF) from the alphaherpesviruses gene UL48 promoter), B112, B42, and the Gal4 activation domain. VP16A showed the strongest activation capacity[19]. To expand the

pool of available elements, we characterised two more TAD modules, a ten-WD-repeat sequence (10*WD)[40] and a Gcn4p *trans*-activating domain (Gcn4 aa107-144; Gcn4$^A$). Mediators are evolutionarily conserved and form the coactivation complex generally required for eukaryotic RNA polymerase II-dependent transcription[41]. Recruiting mediators to upstream activation sequences in promoters may allow transcription, even under inhibitory conditions[42]. To better understanding transactivation effects of mediators, we included the tail mediator 3 (Med3)[42] and the tail mediator 15 (Med15)[42] in the comparison.

We expressed synthetic chimeric TFs consisting of an N-terminal zinc-finger (ZIF) DNA binding domain (Zif268), a linker harbouring a SV40 nuclear localisation sequence (NLS)[43], and TAD (or mediator). VP16$^A$ was included as a reference. We selected the *HAC1* promoter, which delivers moderate and stable expression during both the exponential and ethanol growth phases (Fig. 1c), to control the expression of synthetic TFs (Fig. 1a). This provides a TF expression system that mimics natural expression levels. We used a single-copy centromeric

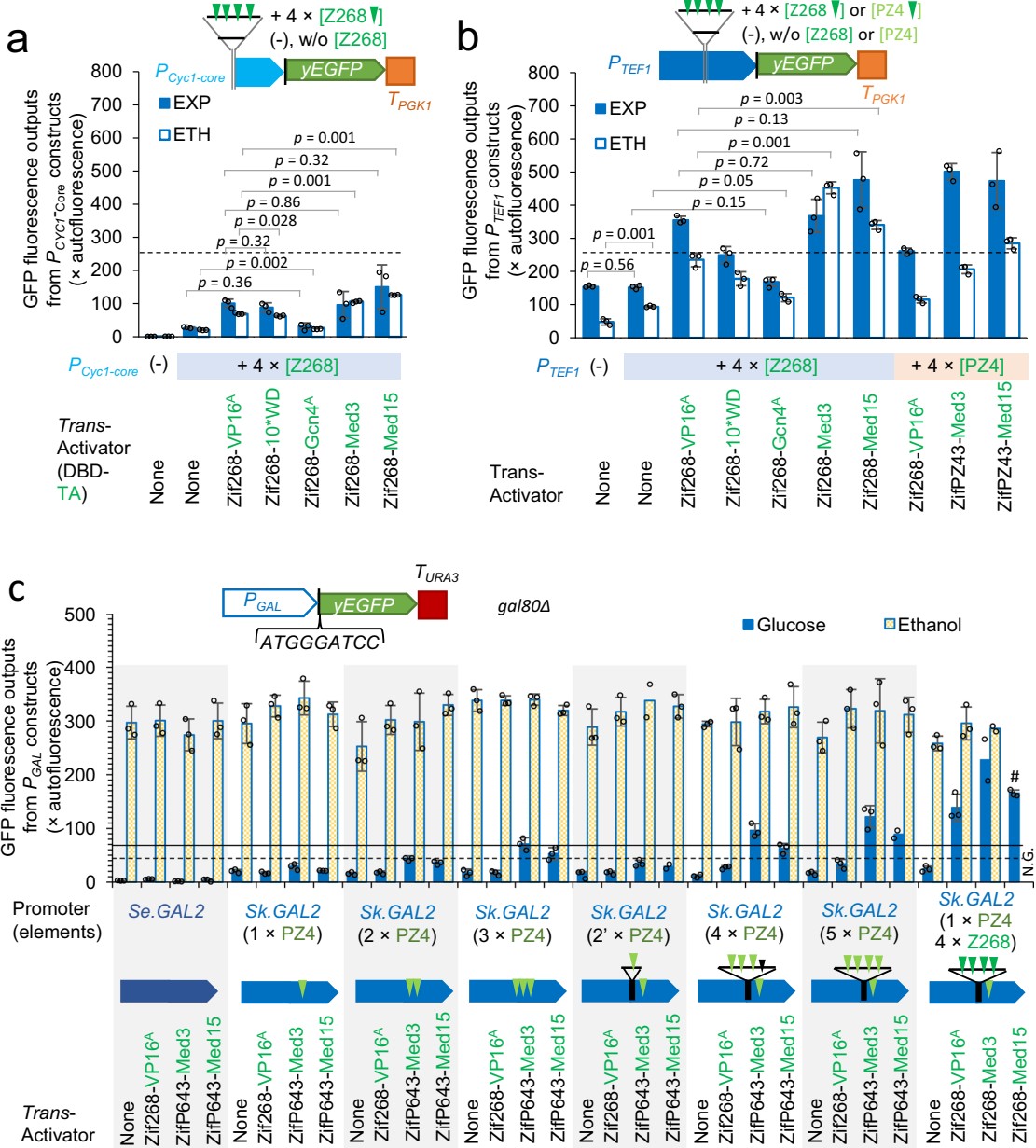

**Fig. 2 Characterisation of *trans*-activation modules. a** The *trans*-activation effects of synthetic *trans*-activators on the hybrid *CYC1* core promoters. DBD, DNA-binding domain. TA, *trans*-activating domain. **b** The *trans*-activation effects of synthetic *trans*-activators on the hybrid *TEF1* promoters. **c** The *trans*-activation effects of synthetic *trans*-activators on the hybrid *GAL* promoters. *Trans*-activators are expressed under the control of the *HAC1* promoter on a single copy centromeric plasmid. The yEGFP fluorescence were characterised in the cells grown to the early exponential growth phase (EXP) or the ethanol-growth phase (ETH) in MES-buffered YNB media using $20\,\text{g}\,\text{L}^{-1}$ glucose as the carbon source in 96-well plates. #, yeast cells were grown in test tubes. N.G., not growing. The dashed horizontal line indicates the output from the native *TEF1* promoter (Fig. 1), and the solid line indicates the output from the native *TDH3* promoter in the EXP-phase cells[39]. GFP fluorescence is expressed as percentage of exponential-phase auto-fluorescence of a GFP-negative strain. Mean values ± standard deviations are shown ($N = 3$ independent biological replicates) or mean values are shown ($N = 3$ independent biological replicates but one replicate was removed as outlier or no data available). Two-tailed Welch's t-test was used for comparing two groups, and p values are shown above the bars. Source data in **a**–**c** are provided in Supplementary Data 1.

plasmid pRS414[44] to introduce TF expression cassettes into yeast. The reference was the empty vector plasmid pRS414.

Synthetic *cis*-regulatory reporter cassettes were constructed using yEGFP as the reporter gene and introduced into yeast via genome integration at the *URA3* locus (Fig. 1a). For a better understanding of *trans*-activation effects of artificial TFs, either the weak core *CYC1* promoter or the strong *TEF1* were fused to a synthetic DNA sequence (truncated from the synthetic P15 promoter, published previously[20]) including four Zif268 binding

elements (Fig. 2a, b; $P_{CYC1+4\times[Z268]}$ and $P_{TEF1+4\times[Z268]}$). Reference promoters were constructed by inserting the synthetic sequence with Z268 elements removed ($P_{CYC1(-)}$ and $P_{TEF1(-)}$).

The *cis*-regulatory reporter cassettes and the *trans*-activator-expressing cassettes were sequentially transformed into the CEN.PK2-1C strain. The *yEGFP* fluorescence from cells in exponential and in ethanol growth phases was measured.

The *CYC1* core promoter, $P_{CYC1(-)}$, showed a very low basal activity (Fig. 2a). Insertion of Z268 elements ($P_{CYC1+4\times[Z268]}$

 

increased basal expression mildly in the absence of an artificial TF. A variety of responses were observed in the presence of synthetic TFs with different TADs. Gcn4[A] showed insignificant activation effects in the exponential growth phase, and a weak effect in the ethanol growth phase. VP16[A], 10*WD, Med3, and Med15 showed >3-fold activation in the cells at the exponential phase. Med3 and Med15 drove significantly higher expression on ethanol than VP16[A].

A short synthetic linker sequence with no biological relevance was into the *TEF1* promoter to provide a landing pad for additional *cis*-acting elements (Fig. 2b). Insertion of this sequence resulted in a decrease in yEGFP expression ($P_{TEF1(-)}$; Fig. 2b). Addition of the Z268 element sequence did not change expression in the exponential phase, but led to a two-fold increase in expression on ethanol (Fig. 2b). Very similar results were observed for expression modification in the presence of the different TAD/mediator-containing constructs (Fig. 2a, b), such that the data from the $P_{TEF1+4\times[Z268]}$ promoter responses positively correlated with the effects observed in the $P_{CYC1+4\times[Z268]}$ promoter (Supplementary Figure 1). The strongest *trans*-activation was observed with Med15, which drove 30–50% higher expression on both exponential and ethanol phases than VP16[A]. Gcn4[A] showed the weakest *trans*-activating capacity, with 10*WD being slightly stronger. VP16[A] and Med3 showed similar activating capacity, in the cells at the exponential growth phase. In contrast to the others, Med3 drove higher expression on ethanol than during the exponential growth phase.

In summary, the artificial *trans*-activators increased the expression from the hybrid *CYC1* promoters by larger fold-changes than that from the hybrid *TEF1* promoters, but because the core CYC1 promoter expression was so weak, the absolute expression increases from the hybrid *TEF1* promoters were larger. Med3 drove an increased activity at the ethanol-growth phase and Med15 had the strongest *trans*-activating capacity. Overall, these results provide insight on the variation of *trans*-activation effects of different protein modules on different core promoters. The stronger activation from Med15 than that from VP16[A] might be because Med15 activates directly, whereas VP16[A] recruits Med15 as a co-mediator to activate[45].

**Additional *trans*-activation modules do not increase the maximal activities of the *Sk.GAL2* promoter.** Previously, we have shown that the *GAL1* promoter is stronger than glycolytic promoters when induced on galactose[39]. We further characterised a set of *GAL* promoters from other *Saccharomyces* species, and found even stronger *GAL* promoters, including the *GAL2* promoters from *S. eubayanus* (*Se.GAL2*) and *S. kudriavzevii* (*Sk.GAL2*)[9]. As shown above, promoter hybrids with *cis*-regulatory elements can be boosted by synthetic *trans*-activators (Fig. 2b). We applied the same mechanism to the *SkGAL2* promoter. The PZ4 *cis*-element 'TA GAG TGA GAC GTT' is found naturally in the *Sk.GAL2* promoter, but not in the *Se.GAL2* promoter or in other *GAL* promoters we have previously characterised[9].

To develop an integrated synthetic *trans*-activating module to boost activity of the *GAL2* promoter, we designed three synthetic Zif domains using ToolGen's Zinc Finger Module set[46] (ZifPZ42 targeting 'GAG TGA GAC GTT', ZifPZ43 targeting 'TGA GAC GTT', and ZifPZ44 targeting 'GAG TGA GAC') and one TALE domain, TALPZ4 (recognising 'TA GAG TGA GAC GTT') based on the dHax3 scaffold[47]. These DNA-binding domains were separately cloned into the N-terminal of the Med3 or Med15 modules to generate synthetic *trans*-activators. Only ZifPZ42 and ZifPZ43 (Fig. 2b)-containing *trans*-activators exhibited *trans*-activation on the hybrid *TEF1-PZ4* promoter ($P_{TEF1+4\times[PZ4]}$), with the ZifPZ43-containing *trans*-activators being stronger.

However, by observing multi-well plate cultures, we saw that the strains expressing ZifPZ43-Med3 and ZifPZ43-Med15 grew slower compared to the reference. Strains expressing ZifPZ42-Med15 and ZifPZ43-Med15 also exhibited a non-homogeneous population (Supplementary Fig. 2). Zif268-Med3 showed a stronger activation on $P_{TEF1+4\times[Z268]}$ on the ethanol growth phase; this was not seen for ZifP643-Med3 with $P_{TEF1+4\times[PZ4]}$ (Fig. 2b). By analysing the sequences inserted into the promoters using the YEASTRACT web-service[48], we found the sequences containing the Z268 and the PZ4 elements also included binding sites for different endogenous yeast TFs (Supplementary Fig. 3). This may indicate a synergistic activation effect of synthetic Zif268-Med3 with endogenous TFs on the hybrid *TEF1* + [Z268] promoter, whereas such effect is not present for ZifPZ43-Med3 on the hybrid *TEF1* + [PZ4] promoter.

We chose the ZifPZ43 *trans*-activator to characterise potential combinatorial effects of additional *trans*-activation components on the *GAL* promoter. We modified the *Sk.GAL2* promoter by inserting one to four additional PZ4 elements, and evaluated the expression outputs from the *Sk.GAL2* promoter and its mutants in the presence of the synthetic *trans*-activators ZifPZ43-Med3 or ZifPZ43-Med15. The *Se.GAL2* promoter (no PZ4 element) was used as a reference. Expression of a yEGFP reporter cassette was examined in the presence and absence of a synthetic *trans*-activator using a *gal80Δ* background strain in early exponential phase on glucose or ethanol (Fig. 2c). Expression of ZifPZ43-Med3 or ZifPZ43-Med15 did not appear to impact on growth rate in these strains.

Despite that *GAL* promoters and the expression of the Gal4 *trans*-activator are repressed by Mig1p in the presence of glucose, basal expression from *GAL* promoters was observed in strains grown on glucose (Fig. 2c). Consistent with previous observations[9], the *Sk.GAL2* promoter showed a higher basal expression on glucose than the *Se.GAL2* promoter. As noted, the *Se.GAL2* promoter does not have a PZ4 element, and expression of *trans*-activators ZifPZ43-Med3 or ZifPZ43-Med15 did not increase yEGFP expression from the *Se.GAL2* promoter. However, expression of Zif268-VP16[A] led to a two-fold increase of yEGFP fluorescence, despite that the *Se.GAL2* promoter does not contain Z268 element. Nevertheless, GFP fluorescence was still very low in this strain on glucose.

For strains harbouring the *Sk.GAL2* promoter and its PZ4-element mutants growing on glucose, activators ZifPZ43-Med3 and ZifPZ43-Med15 increased yEGFP expression (Fig. 2c). There was a positive correlation between the number of PZ4 elements and the observed GFP fluorescence (Supplementary Fig. 4). In contrast to our previous findings with the *TEF1* promoter (Fig. 2b), the Med3 mediator drove a stronger activation than the Med15 mediator. Zif268-VP16[A] drove a ~2-fold increase in the expression from *Sk.GAL2* 4×[PZ4] promoter and *Sk.GAL2* 5×[PZ4] promoter (Fig. 2c; but not from other 'PZ4'-only *Sk.GAL2* promoters). This may indicate that there is a non-specific *trans*-interaction between Zif268 and the sequences inserted into *Sk.GAL2* 4×PZ4 promoter and *Sk.GAL2* 5×[PZ4] promoter. The *Sk.GAL2* 5×[PZ4] promoter in the presence of ZifPZ43-Med3 showed ~3-fold stronger expression output, in comparison with the commonly-used *TEF1* promoter with the same yEGFP-reporter construct (Construct 1, Fig. 1), and 1.7-fold stronger than the *TDH3* promoter[39]. This shows the strong *trans*-activation effects from the artificial *trans*-activator-*cis* element interaction on the hybrid *GAL* promoter on glucose.

On ethanol, the *Se.GAL2*, *Sk.GAL2*, and *Sk.GAL2* + [PZ4] promoters were activated to a very high level (Fig. 2c), at least seven-fold higher than the *TEF1* promoter (Construct 1, Fig. 1). Neither ZifPZ43-Med3 nor ZifPZ43-Med15 significantly *trans*-activated the *Sk.GAL2* + PZ4 promoters on ethanol.

 

In addition to the $Sk.GAL2 + $ [PZ4] promoters, we also constructed a $Sk.GAL2 + $ Z268 hybrid promoter, and tested the trans-activation effects of artificial trans-activators Zif268-VP16[A], Zif268-Med3, and Zif268-Med15. Expression of these trans-activators increased the yEGFP activity from the $Sk.GAL2 + $ [Z268] promoter by at least five-fold (Fig. 2c) on glucose. Zif268-Med3 showed the strongest activation effect, resulting in the expression of $Sk.GAL2 + $ Z268 promoter on glucose being near to the levels on ethanol. These findings further confirmed the strong trans-activation capacity of artificial trans-activators.

However, by observing the cultures in multi-well plates, we saw that $Sk.GAL2 + $ Z268 promoter-testing strain (gal80Δ background) with Zif268-Med3 and Zif268-Med15 grew slower compared to the reference, and the Zif268-Med15-expressing strain did not grow on ethanol. Such effects were not seen in the $TEF1/CYC1 + $ [Z268] promoter-testing strains ($GAL80$-wildtype; Fig. 2). We postulated that the growth impairment might be caused by untargeted trans-activation from Zif268-Med3 and Zif268-Med15 in gal80Δ background.

In summary, artificial trans-activators can significantly activate the $GAL$ promoter via the orthogonal cis-elements on glucose. However, they did not further increase the activities of the $GAL$ promoters on ethanol, indicating that these activators do not significantly increase its maximum expression output of the $GAL$ promoter.

**Eukaryote-like trans-repression mechanisms can be used to build effective repression circuits in yeast.** Trans-repression is another natural mechanism for transcriptional regulation. In bacteria, a repressor commonly functions by binding to the elements near the transcription start site to prevent transcription from the target promoter[49,50]. A similar mechanism can be reconstructed in yeast by the insertion of bacterial repressor-binding sites near to 'TATA-box' or transcription initiation sequence in the target promoter and the expression of the cognate bacterial repressor[6,15,21]. However, intensive effort, including construction and screening of large libraries, was required to obtain the ideal hybrid repressible promoter; in addition, a strong constitutive promoter is required to express bacterial repressors[6,15]. In contrast to bacterial repressors, transcriptional repressors in eukaryotes commonly mediate gene repression through a series of regulatory mechanisms including recruiting histone deacetylase complexes to direct nucleosome formation and blocking co-activators[51,52]. These mechanisms may be more relaxed about the position of cis-acting elements in target promoters and can potentially be exploited to develop a trans-repressing toolset.

We characterised the trans-repressing capacity of four protein modules, including Sin3 C-terminal domain (Sin3 aa600-1536; Sin3[C])[53], Tup1[54], Cyc8[54], and Mig1 C-terminal domain (Mig1 aa123-504; removal of DNA-binding domain; Mig1[C])[55]. Sin3[C] mediates repression through its interactions with histone deacetylase complexes[53], Tup1/Cyc8 through recruiting histone deacetylase complexes and masking binding of co-activators[54], and Mig1[C] is thought to recruit the Tup1/Cyc8 complex[55]. Sin3[C], Tup1, and Cyc8 have been previously tested in various studies by fusing them to a DNA-binding domain, but Mig1[C] has not[53,54].

We first attempted to characterise the fusion of Zif268-SV40[NLS] with the C-terminal Sin3[C], Tup1, or Mig1[C]. The $HAC1$ promoter was used to control the fusion expression. yEGFP controlled by $P_{TEF1+4\times[Z268]}$ was used as the reporter (Fig. 3a). However, yeast transformants of Zif268-Sin3[C] and Zif268-Tup1 constructs grew extremely slowly on agar plates, and the colonies could not be picked after more than a week of incubation. We

were able to characterise the strain expressing Zif268-Mig1[C]. We observed two distinct cell populations: in one population, the yEGFP expression was severely repressed, whereas in the other, no repression (relative to the control construct) was observed (Supplementary Fig. 5).

The unstable expression profiles in the Zif268-containing promoter constructs suggested off-target effects rendering the constructs unusable. We therefore developed an alternative approach using TetR elements as the DNA-binding domain for characterisation of the four trans-repressing modules. A cis-regulatory reporter cassette was constructed by replacing the Z268 elements in $P_{TEF1+4\times[Z268]}$ with TetO elements. The control construct in the absence of a repressor module showed ~2-fold higher expression in the exponential phase than in the ethanol phase and did not respond to tetracycline addition (Fig. 3a).

Yeast expressing TetR-derivative repressors (TetR-Sin3[C], TetR-Tup1, TetR-Cyc8, and TetR-Mig1[C]) grew normally on plates and showed strong repression on the $TEF1 + 4\times$[TetO] promoter in the absence of tetracycline. TetR-Tup1 was most efficient, showing >100-fold repression and eliminating yEGFP fluorescence in > 90% cell population (Fig. 3a; Supplementary Fig. 5). TetR-Cyc8, and TetR-Mig1[C] showed the repression similar to TetR-Tup1. The repression was maintained in the cells at the ethanol-growth phase, even for the cells expressing TetR-Mig1[C], although Mig1 itself is a glucose-dependent repressor. Repression on ethanol by TetR-Mig1[C] might be due to the inclusion of a nuclear localisation sequence (Fig. 1a), which dominates the glucose-dependent nuclear localisation of Mig1[56]. Addition 50 μM tetracycline can de-repress yEGFP expression in the whole cell population (Supplementary Fig. 6). TetR-Sin3[C] was less efficient, not eliminating yEGFP fluorescence in the whole population (Fig. 3a; Supplementary Fig. 6). Yeast expressing TetR-repressors did not show a homogenous population: ~ 8% population deviated from the major population and showed higher yEGFP fluorescent levels under the conditions without tetracycline additions (TetR-Tup1 in Supplementary Fig. 5). We postulated that this might be caused by the loss of repressor-expressing plasmids: centromeric plasmids show a 3% loss rate per generation[57].

The repression was maintained in the cells at the ethanol-growth phase, even for the cells expressing TetR-Mig1[C], although Mig1 itself is a glucose-dependent repressor. Repression on ethanol by TetR-Mig1[C] might be due to the inclusion of a nuclear localisation sequence (Fig. 1a), which dominates the glucose-dependent nuclear localisation of Mig1[56].

In summary, repression capacities of the four trans-repressing domains were sequenced as Tup1 ~ Mig1[C] ~ Cyc8 » Sin3[C], and a low-level expression of a TetR fusion with Tup1, Mig1C, or Cyc8 repressor can fully repress the expression of a strong promoter inserted with TetO elements.

**Tetracycline-mediated repression of $GAL$ promoters.** To develop a mechanism that can fully repress $GAL$ promoters-controlled metabolic pathways on demand, we used the tetracycline de-repressible circuit to control the expression of the $GAL$ repressor Gal80p. In the resulting strain, TetR-Tup1 was expressed under the control of the $HAC1$ promoter, the $TEF1 + $ [TetO] promoter was integrated upstream of the $GAL80$ gene to control its expression, and a yEGFP gene was introduced under the control of the $GAL1$ promoter to report the regulatory effects of tetracycline (Fig. 3b). The strain was first grown on ethanol to relieve Mig1p-mediated glucose repression on $GAL$ promoters[2,9], in the absence or presence of tetracycline. In the absence of tetracycline, a very high level of yEGFP fluorescence was detected (Fig. 3c and Supplementary Fig. 6), showing a fully

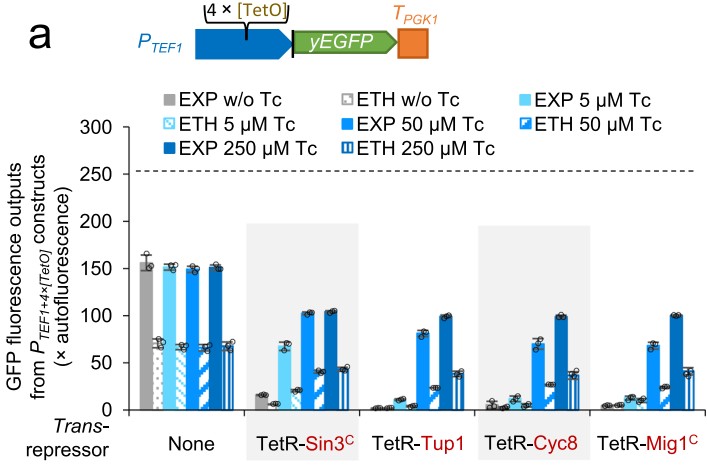

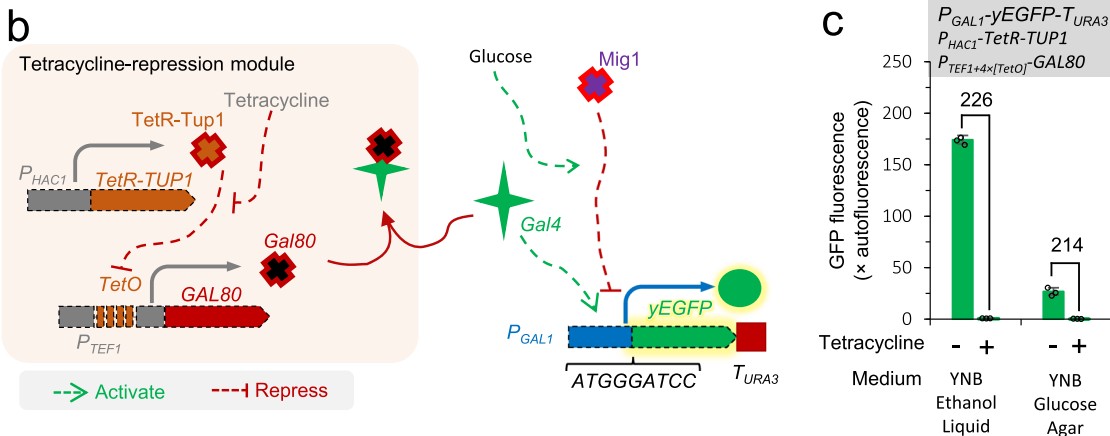

**Fig. 3 Characterisation of *trans*-repression modules. a** The *trans*-regulating effects of TetR-repressors with or without tetracycline addition. The yEGFP fluorescence were characterised in the cells grown to the early exponential growth phase (EXP) or the ethanol-growth phase (ETH) in MES-buffered YNB media using 20 g L$^{-1}$ glucose as the carbon source in 96-well plates. The dashed horizontal line indicates the output from the native *TEF1* promoter (Fig. 1). **b** Schematic of tetracycline-mediated repression on the *GAL* promoter. **c** The effects of tetracycline on the *GAL* promoter in the cells integrated with tetracycline-repression module in **b**. YNB Ethanol Liquid: yeast cells were grown to OD600 ~ 1 in MES-buffered YNB media using 2% (v/v) ethanol as the carbon source at 30 °C in testing tubes and analysed. YNB Glucose Agar: yeast was streaked on YNB nutrient agar with 20 g L$^{-1}$ glucose as the carbon source at 30 °C for 48 h, and single colonies were resuspended in water and analysed. The numbers on top of bars are the decrease fold-change at 37 °C to 30 °C. Mean values ± standard deviations are shown (N = 3 independent biological replicates). Source data in **a** & **c** are provided in Supplementary Data 1.

de-repression state of the *GAL1* promoter[2,39]. In the presence of 125 μM tetracycline, yEGFP fluorescence was repressed by >200 fold to very minimal fluorescence.

We then grew the strain on nutrient agar containing 20 g L$^{-1}$ glucose as the carbon source. Consistent with our previous observation that 20 g L$^{-1}$ glucose in agar is not sufficient to repress the expression from *GAL* promoters[9], the *GAL1* promoter was moderately de-repressed in the absence of tetracycline (Fig. 3c and Supplementary Fig. 6). Addition of 125 μM tetracycline repressed the yEGFP expression by >200 fold.

In summary, by integrating tetracycline-inducible expression circuit to control *GAL80* expression, we successfully deployed a tetracycline-mediated mechanism to repress *GAL* promoters.

**A heat-mediated response circuit for 37 °C induction of *GAL* promoters**. Temperature is a primary parameter to control during yeast cultivation. Although the preferred temperature for *S. cerevisiae* is 30 °C, growth at higher temperature can potentially reduce the cooling costs for exothermic fermentation processes[58], and increased temperatures are well tolerated. Transition to an increased bioprocess temperature upon entry into the production phase (when fast growth is no longer required) would therefore

make a convenient tool for regulatory control and may make the bioprocess more economical. To exploit this potential, we designed a heat-inducible circuit to modulate expression from *GAL* promoters in strains growing on glucose. Our model employs a heat-inducible degron[59] to regulate the stability of the glucose-dependent repressor Mig1p (Fig. 4b).

We first used the yEGFP reporter to characterise the previously-developed heat-inducible degron (H.degron)[59] in our system. The full heat-inducible degron module comprises four parts: a ubiquitin moiety (Ubi) that is cleaved off by ubiquitin C-terminal hydrolase after translation, an 'N-end rule' residue Arg, a linker 'HGSGTMV', and mouse dihydrofolate reductase P66L temperature sensitive mutant (DHFR*). The heat-inducible degron Ubi4-RHGSGTMV-DHFR* was fused to the N-terminus of yEGFP, and the fusion (UBI4-RHGSGTMV-DHFR*-yEGFP) was expressed under the control of *TEF1* promoter and the *PGK1* terminator (Fig. 4a). yEGFP was characterised as the control. Two resulting yeast strains, expressing yEGFP and UBI4-RHGSGTMV-DHFR*-yEGFP, were grown at 30 °C or at 37 °C. Consistent with previous studies[59,60], elevation of growth temperature to 37 °C did not slow the growth. In the yEGFP control, fluorescence was 1.2-fold lower at 37 °C (Fig. 4c),

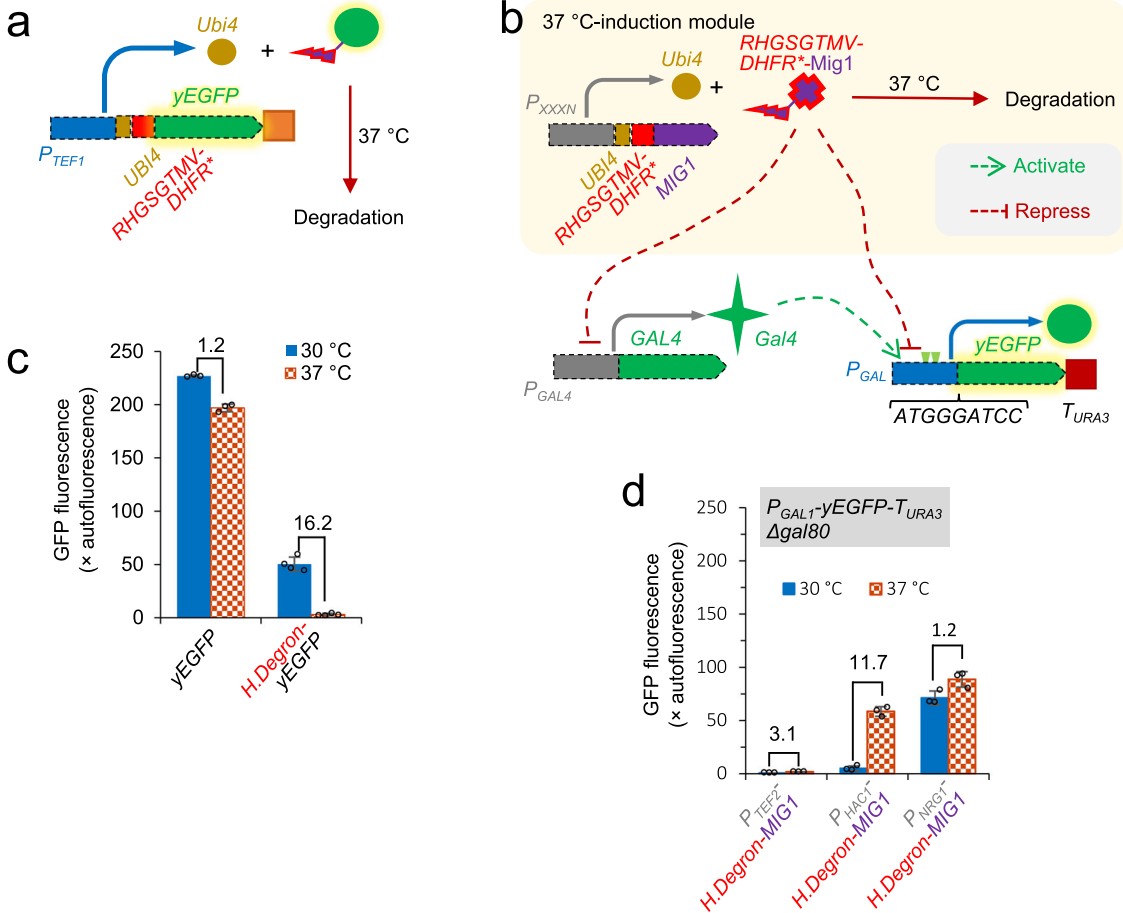

**Fig. 4 Deploying 37 °C *GAL*-induction circuit. a** Schematic of expression system of heat-inducible degradation of yEGFP. **b** Schematic of genetic circuits of 37 °C-induction of the *GAL* promoter. **c** Degradation efficiency of heat-inducible degron in **a** at 37 °C. **d** Induction response of the *GAL* promoter at 37 °C. H. Degron, Ubi4-RHGSGTMV-DHFR*. Yeast cells were grown to the early exponential growth phase in MES-buffered YNB media with 20 g L$^{-1}$ glucose as the carbon source in testing tubes. The numbers on top of bars are the decrease fold-change at 37 °C to 30 °C. Mean values ± standard deviations are shown ($N = 3$ or 4 independent biological replicates). Source data in **c** and **d** are provided in Supplementary Data 1.

indicating that elevated temperature did not dramatically affect yEGFP abundance.

Tagging Ubi4-RHGSGTMV-DHFR* decreased fluorescence at 30 °C by 4.5-fold compared to the control yEGFP (Fig. 4c), suggesting either disruption of transcription/translation or basal degradation by the degron. At 37 °C, fluorescence decreased by 16.2-fold. These data demonstrate the functionality of Ubi4-RHGSGTMV-DHFR* as an effective heat-inducible degron (H.Degron).

In summary, the basal degradation (or impaired expression) at 30 °C caused by Ubi4-RHGSGTMV-DHFR* (H.Degron) is significant; moreover, heat-inducible degradation by this degron at 37 °C, although dramatically decreasing yEGFP abundance, did not result in a full depletion of target yEGFP (Fig. 4c). Incomplete degradation might be problematic for application to control the abundance of a transcription factor, because transcription factors can function effectively at a relatively low expression level. To apply the H.Degron to regulate the abundance of Mig1p for the proposed 37 °C-mediated de-repression of *GAL* promoters, it will be important to overcome these two limitations, otherwise a sharp temperature-induced boundary effect cannot be achieved.

To deploy the 37 °C-mediated de-repression module on *GAL* promoters, we used a *gal80Δ* strain expressing *yEGFP* under the control of the *GAL1* promoter and the *URA3* terminator. In this strain, we modified the *MIG1* locus by fusing the H.Degron to the N-terminus of Mig1p and introducing an alternative promoter to

control the expression strength. We tested three promoters to provide variable expression strength with the aim of titrating the transcription of *H.Degron-MIG1* to obtain a sharp boundary effect: the *TEF2* promoter (strong)[39], the *HAC1* promoter (medium-weak; Fig. 1c), and the *NRG1* promoter (weak; Fig. 1c). The resulting yeast strains were grown on glucose at 30 °C or 37 °C to examine yEGFP expression from the *GAL1* promoter (Fig. 4d). In these cases, cultivation at 37 °C did not result in slowed growth. The strain with the *TEF2* promoter construct showed a very low level of yEGFP fluorescence at 37 °C, indicating the expression from the *TEF2* promoter was too high and the H.Degron was not sufficient to deplete Mig1p. The strain with the *NRG2* promoter showed an induced level of yEGFP fluorescence at 30 °C, indicating that the expression from the *NRG2* promoter was not sufficient and the basal degradation of H.Degron caused the depletion of Mig1p. Fortunately, the strain with the *HAC1* promoter construct showed very low level yEGFP fluorescence at 30 °C and good induction 37 °C (Fig. 4d and Supplementary Fig. 6). This confirmed the functionality of 37 °C-mediated de-repression of *GAL* promoters.

**Expanded regulatory circuits for conditional control of sesquiterpene (nerolidol) production.** Previously, we developed yeast strains for high-level sesquiterpene production through metabolic engineering[17,61,62]. To develop these strains, *GAL*

promoters, coupled with the deletion of *GAL* repressor gene *GAL80*, were used to control the expression of the genes from the mevalonate pathway and the genes for production of the sesquiterpene *trans*-nerolidol. The resulting strain NLD401 produced ~1.7 g L$^{-1}$ nerolidol in two-phase flask cultivation at 30 °C[17]. In the current work, we developed an isogenic yeast strain (N9R5HIRBU), in which the synthetic genetic modules for tetracycline-mediated repression and 37 °C-mediated de-repression of *GAL* promoters were also established (Fig. 5a). In this strain, Yellow Fluorescence-Activating and absorption-Shifting Tag (YFAST) and *Actinidia chinensis* (kiwifruit) nerolidol synthase (AcNES1) were expressed through a 2A sequence-mediated bicistronic expression mechanism (*YFAST-2A-AcNES1*)[17]. The *YFAST-2A-AcNES1* was under the control of the *GAL2* promoter. This construct not only contributed to an improved nerolidol production, but also allowed us to monitor the *GAL2* promoter activities by measuring YFAST fluorescence throughout the whole cultivation as a proxy for *GAL2* promoter-driven expression. We evaluated the effects of tetracycline and elevated temperature on nerolidol production and the *GAL* promoter activities in this strain via two-phase dodecane-overlayed flask cultivation.

In flask cultivation at 30 °C and the absence of tetracycline, strain N9R5HIRBU (+ tetracycline-repressible/heat-inducible regulatory modules: Fig. 3b and Fig. 4b) showed a specific maximum growth rate ($\mu_{max}$) of 0.3 h$^{-1}$ and a maximum cell density at OD$_{600}$ of ~14 at 72 hr. N9R5HIRBU produced ~1.76 g L$^{-1}$ nerolidol at 72 hr (Fig. 5e). The dynamics of YFAST fluorescence were also similar in both strains: a very low expression in the first 12 hr when glucose was the carbon source and a gradually induced expression after 12 hr when glucose was depleted (Fig. 5c). These physiological features were similar to the isogeneic strain NLD401 published previously (w/o tetracycline-repressible/heat-inducible regulatory modules; *gal80Δ*)[17]. These data show that the 'OFF' state of tetracycline-mediated repression on *GAL* promoters is tight, producing a *gal80Δ*-like genotype and phenotype.

Addition of 125 μM tetracycline at 30 °C did not significantly alter growth characteristics in the exponential phase (Fig. 5b and d), however much more biomass was produced during the post-exponential phase, resulting in a maximum OD$_{600}$ of ~22 (Fig. 5b). Consistent with tetracycline-mediated repression on yEGFP expression from the *GAL1* promoter (Fig. 3), YFAST expression in N9R5HIRBU was essentially fully repressed by 125 μM tetracycline throughout the whole cultivation (Fig. 5c). Only ~0.03 g L$^{-1}$ nerolidol was produced at 72 hr, 60-fold lower than in the absence of tetracycline at 30 °C (Fig. 5e). These data demonstrate that the 'ON' state of tetracycline-mediated repression on *GAL* promoters provided a fine mechanism for nearly full repression on *GAL* promoter-controlled heterologous synthetic pathways. It also demonstrates that nerolidol production delivers a metabolic burden which decreases biomass production. The tetracycline repression module can facilitate strain maintenance during laboratory manipulation by relieving the metabolic burden caused by heterologous synthetic pathways.

We then characterised strain N9R5HIRBU at 37 °C in the absence of tetracycline to investigate the effects of elevated temperature on nerolidol production. Pre-culturing was performed at 37 °C to prepare the seed inoculum. A decrease in exponential growth rate and biomass accumulation was observed at 37 °C, with a maximum OD$_{600}$ of ~11 (Fig. 5b and d). YFAST fluorescence at 0 h was ~20-fold higher at 37 °C than at 30 °C, and was continuously induced during the exponential and post-exponential growth phases (Fig. 5c). This suggests that heat-inducible protein degradation might not be efficient for complete depletion of Mig1p, but does provide a moderate relief of glucose repression. Reflecting the induced expression from the *GAL*

promoter, strain N9R5HIRBU produced ~ 0.5 g L$^{-1}$ nerolidol at 24 h and 37 °C, ~6-fold improvement compared to that at 30 °C; and produced ~2.54 g L$^{-1}$ nerolidol at 72 h, a 45% improvement on the original NLD401 strain and on the N9R5HIRBU strain grown at 30 °C (Fig. 5e). Consistent with this, the specific nerolidol production rates in the cells grown at 37 °C were significantly higher than those in the cells grown at 30 °C, with a 9-fold increase in the first 24 h and a 49% increase from 24 hr to 72 h (Fig. 5f). These data demonstrate that the integration of 37 °C-mediated induction circuits for the *GAL* promoters resulted in improved production of nerolidol-producing strains at an elevated temperature.

## Discussion

Genetic regulation tools should be applicable in biotechnology for development of microbial cell factories in metabolic engineering[63,64]. Such tools can be used in industrial processes to dynamically tune host cell metabolism for maximal productivities, or in the laboratory to facilitate strain maintenance and engineering. Development of these tools is often encumbered by unknown properties of basic genetic modules under variable conditions. Therefore, in this work, we started with characterisation of basic genetic circuits, including eukaryote-like *trans*-activating and *trans*-repressing modules, and then deployed new mechanisms to regulate *GAL* promoters aiming for application in metabolic engineering. Our approaches included: characterisation of different promoters from transcription factor genes across exponential and ethanol phases to better understand endogenous eukaryotic controls on regulatory proteins (Fig. 1), engineering promoters with *cis*-elements in combination with *trans*-acting factors to modulate transcriptional activity to constitutive and galactose-responsive promoters (Fig. 2), establishing repression circuits based on eukaryotic systems (Fig. 3), establishing a temperature-inducible circuit (Fig. 4), and applying selected circuits to production of nerolidol (Fig. 5).

In terms of basic promoter-mediated control, we observed that promoters from transcription factor genes expressed at 1-2 orders of magnitude lower than commonly used constitutive promoters. We used this information to identify preferred promoters for control of transcription factors in engineered circuits, so that regulation similar to endogenous systems could be achieved. We then applied this to regulate *cis*-element engineered promoters. We found that *trans*-activating modules have variable effects on transcriptional activity from different hybrid promoters; in particular, the weak *CYC1* promoter demonstrated a much higher fold-change in response to *trans*-activating module engineering than the strong *TEF1* promoter (Fig. 2). Our data suggest that (a) the activation effects of an artificial *trans*-activator varies depending on the core promoter used in the reporter system, and (b) threshold effects exist for *trans*-activating module activity in engineered promoters. In support of this, while there was generally a positive correlation between the number of *cis*-acting elements inserted in engineered promoters, additional *trans*-activation modules did not increase the maximal activities of the *Sk.GAL2* promoter, which can already be activated to very high levels (Fig. 2c and supplementary Fig. 4). We also observed a potential syngenetic effect with endogenous TFs, shown by Zif268-Med3 on *TEF1* + [Z268] promoter (Fig. 2b). Overall, these observations are consistent with the previous observation that core promoter sequence plays a major role determining expression level[65]. They further suggest that maximal activation by Gal4p on the *GAL* promoter under de-repression conditions had been achieved, and suggested that any further increases would require engineering of the core promoter.

Moreover, some synthetic TFs like Zif268-Tup1 and Zif268-Sin3$^C$ are not compatible with yeast, leading to the problems or

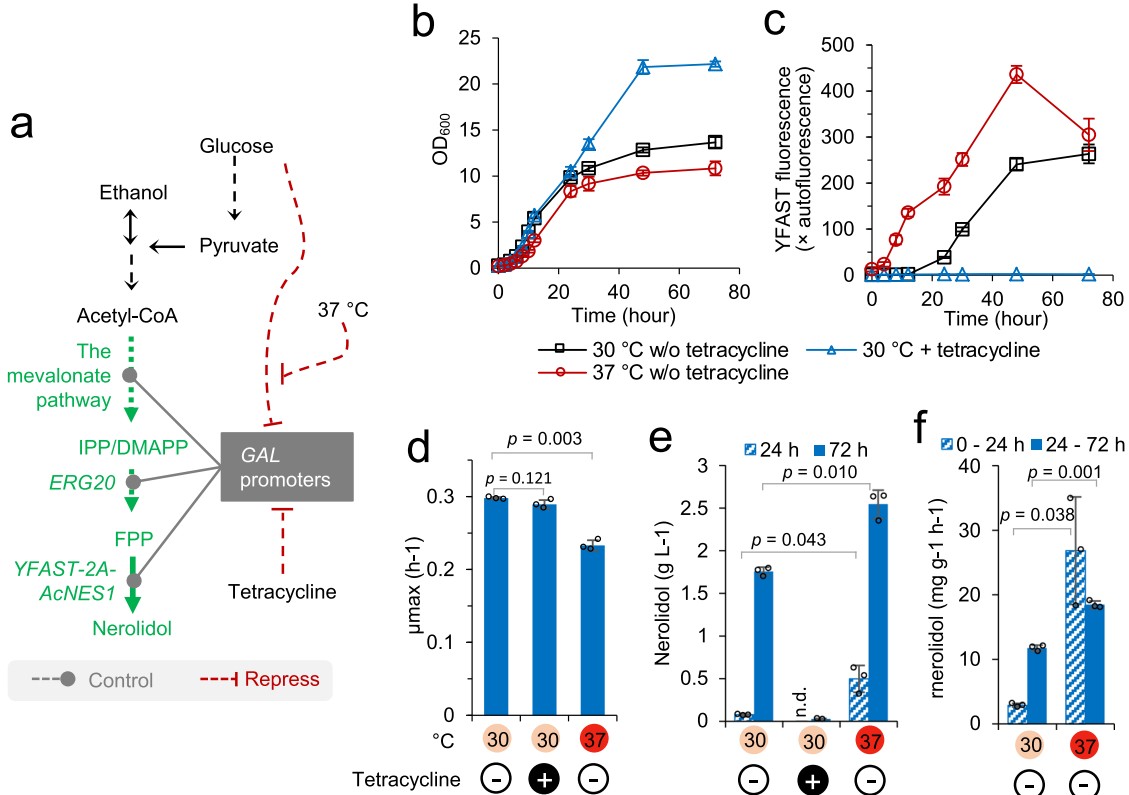

**Fig. 5 Characterisation of nerolidol-producing yeast (strain N9R5MIRBU) with *GAL* promoters controlled via tetracycline-repressible and heat-inducible mechanisms. a** Schematic of nerolidol synthetic pathway and *GAL* promoter regulation in strain N9R5MIRBU. Tetracycline/heat regulatory mechanisms on *GAL* promoters refer to Figs. 3b and 4b. IPP, isopentenyl pyrophosphate; DMAPP, dimethylallyl pyrophosphate; FPP, farnesyl pyrophosphate; *ERG20*, FPP synthase; *YFAST-2A-AcNES1*, 2A-mediated bicistronic gene of Yellow Fluorescence-Activating and absorption-Shifting Tag (*YFAST*) and *Actinidia chinensis* (kiwifruit) nerolidol synthase (*AcNES1*). **b–f** Characterisation of strain N9R5MIRBU in two-phase flask cultivations under the conditions: at 30 °C and in the absence of tetracycline (30 °C w/o tetracycline); at 30 °C and in the presence of 125 μM tetracycline (30 °C + tetracycline); or at 37 °C and in the absence of tetracycline (37 °C w/o tetracycline). The yeast cells were grown in 2-(N-morpholino)ethanesulfonic acid (MES)-buffered synthetic mineral salt medium (prepared from yeast nitrogen base, YNB) with 20 g L$^{-1}$ glucose as the carbon source. YFAST fluorescence was measured after 4-hydroxy-3-methylbenzylidene rhodanine (HMBR) with final concentration 20 μM was added in yeast samples before flow cytometry assay. Mean values ± standard deviations are shown (*N* = 3 independent biological replicates) in **b–d**, **e** for 30 °C w/o tetracycline and 37 °C w/o tetracycline, and **f**. Mean value and absolute errors are shown in **e** for 30 °C + tetracycline (*N* = 2 independent biological replicates). n.d., not determined. Source data in **b–f** are provided in Supplementary Data 1.

failure in transformation, whereas expression of some TFs like Zif268-Med3 and Zif268-Med15 in *gal80Δ* background, but not in *GAL80* background, decreased growth fitness. Similarly, toxicity of artificial TFs like PhlF-VP16[A], CamR-VP16[A], lexA-VP16[A] was previously reported[19,33]. Toxicities associated with Zif nucleases are known to be caused by off-target effects[66]. Similarly, such detrimental effects of these artificial TFs might be caused by off-target interactions that perturb genetic networks for growth fitness. It is noteworthy that Zif268-Med3/15 toxicity is apparently masked by the Gal80 repressor. Genetic profiling will be required to identify shared targets that result in toxic effects. Also, Zif268-VP16[A], although summoning Med15 for *trans*-activation[45], did not show off-target toxicity (which would be evidenced by growth defects) in the *gal80Δ* background. Similarly, Zif268-Mig1[C], but not Zif268-Tup1, could be transformed into yeast, although Mig1[C] recruits the Tup1-Cyc8 repression complex[55]. These results suggest that adding a layer of protein-protein interaction may provide an equilibrium to the system, resulting in attenuation of off-target toxicity. On the other hand, such a multi-layer regulatory mechanism may contribute to the specificity of transcription factor regulation[67].

Adding to the component toolbox, we went on to establish repression circuits based on eukaryotic *trans*-repressing modules,

similarly to the approach used for the *trans*-activating modules (*cis*-element engineering on promoters with concurrent expression of *trans*-repressing transcription factors). This system was established using the *TEF1* promoter; the established module was then applied to control the *GAL80* repressor, providing a further level of control to the 'on' and 'off' switch and decrease leakiness of the *GAL* promoter.

For application in yeast metabolic engineering, we successfully deployed the circuit for tetracycline-mediated repression on *GAL* promoters and the circuit for 37 °C-mediated induction of *GAL* promoters. Improving on previous tetracycline regulatory circuits[15,18,33], the TetR-derivative circuit developed in this study fully represses the *TEF1* + [TetO] promoter in absence of tetracycline (Fig. 3a) and a weak expression of TetR-fusion repressor is efficient for such repression. Moreover, we used it to control *GAL80* expression delivering an efficient binary ON/OFF control on *GAL* promoters (Fig. 3b, c). In contrast, the heat-inducible degron in isolation was not efficient to deplete the target protein. In this case, precise titration of the expression strength of Mig1p fused with heat-inducible degron was required to deliver a binary ON/OFF effect on *GAL* promoters (Fig. 4d). Combination of these two circuits provides novel operation principles: (1) heterologous pathways can be repressed in small scale laboratory

development procedures by addition of a small molecule, which facilitates strain maintenance, rather than adding the molecule to induce these pathways in industrial processes; (2) heat released from large-scale fermentation can be hijacked for metabolic pathway induction, also decreasing the overall bioprocess cost. Although not fully de-repressing the *GAL* promoter on glucose (Fig. 5c), heat-inducible degradation of Mig1p led to improved nerolidol productivities at 37 °C (Fig. 5). This prototyped a process in *S. cerevisiae* featuring independence from expensive inducers and improvement in production. However, further validation is necessary for its application in real industrial processes, and to determine if this approach can reduce cooling costs.

In summary, proper characterisation of each modularised genetic module prior to deploying them in a circuit format was an essential element for success of this project. Although not all characterised genetic modules were eventually applied in development of metabolic engineering tools, such characterisation revealed basic features of these modules, which provides guidance for circuit design. Continuing to build up part libraries and even re-characterisation of known parts under varied conditions and/ or through independent systems or procedure will improve our knowledge about their features and is important for precise synthetic circuit design. Capacities for biofoundry-based molecular laboratories[68,69], model-driven design[70], and directed evolution of module components[33] will be important for symmetrical and modularised engineering for synthetic circuits. Fortunately, the two novel regulatory models were straightforward to develop, and are generally applicable in metabolic engineering. Isolated modules, either tetracycline-mediated repression or 37 °C-mediated induction of *GAL* promoters, can be integrated in previously developed strains where *GAL* promoters are used to control the expression of heterologous metabolic pathways. It should be also emphasised that synthetic regulatory circuits for metabolic engineering can be engineered towards both facilitating laboratory strain maintenance and bringing economic mechanisms for pathway control during production.

## Methods

**Plasmid and strain construction**. Plasmids used in this work are listed in Supplementary Table 1, and strains are listed in Supplementary Table 2. Primers used in polymerase chain reaction (PCR) and PCR performed in this work are listed in Supplementary Table 3. Plasmid construction processes are listed in Supplementary Table 4. Yeast strain construction processes are listed in Supplementary Table 5. A LiAc/SS carrier DNA/PEG method[71] was used for yeast transformation. Molecular Cloning Designer Simulator was used to manage DNA sequence design[72]. Promoters inserted with synthetic cis-elements and synthetic protein domains are listed in Supplementary Table 6.

**Yeast cultivation**. Yeast nitrogen base without amino acids (YNB, FOR-MEDIUM#CYN0402; 6.9 g L$^{-1}$) was used to prepare the media with ammonium as the nitrogen source. When urea was used as the nitrogen source, 1.7 g L$^{-1}$ yeast nitrogen base without amino acids and without ammonium sulphate (Sigma Aldrich) and 1 g L$^{-1}$ urea were used to prepare the base media, and sterilised by filtration. Amino acids (leucine, histamine, tryptophan) were supplemented in YNB media to grow auxotrophic strains. For the YNB media without additional buffer, pH was adjusted to 6.0 using sodium hydroxide solution; for the YNB media with 2-(N-morpholino)ethanesulfonic acid (MES) buffer, 19.5 g L$^{-1}$ MES was used, and pH was adjusted to 6.0 with ammonia hydroxide solution. Glucose (20 g L$^{-1}$) or ethanol (2% v/v) was used as the carbon source.

For characterisation of yEGFP-expressing strains, yeast cells from glycerol stocks were streaked on YNB-glucose agar. For the growth in 96-well microplates, yeast cells were grown in YNB-glucose medium for about 20 h to stationary phase in a 350 rpm 30 °C incubator to prepare seed culture. Seed culture (5 µl) was inoculated into 100 µl MES-buffered YNB-glucose medium to prepare Culture 1. Culture 1 (2 µl) was inoculated into 100 µl MES-buffered YNB-glucose medium to prepare Culture 2. Culture 2 was incubated in a 350 rpm 30 °C incubator overnight for analysis of yEGFP fluorescent in the cells grown to the exponential growth phase, and Culture 1 for ~36 h for analysis in the cells grown to the ethanol growth phase. Seed culture (2 µl) was inoculated into 100 µl YNB-ethanol medium, and the culture was incubated in a 350 rpm 30 °C incubator for ~48 h for analysis in the cells grown grow on ethanol. For the growth in 10 ml test tubes, yeast cells were

inoculated into 500 µl MES-buffered YNB-glucose medium and grown overnight in a 200 rpm 30 °C or 37 °C incubator for analysis of yEGFP fluorescent in the cells grown to the exponential growth phase. Stock tetracycline solutions (125 mM) was prepared in DMSO-ethanol (1:1) solution and supplemented to prepare the medium with 125 µM or 250 µM tetracycline, and 25 mM stock solution for the medium with 50 µM tetracycline, and 2.5 mM stock solution for the medium with 5 µM tetracycline.

For characterisation of nerolidol-producing strains, dodecane-overlayed two-phase flask cultivation was used. Yeast cells from glycerol stocks were streaked on YNB -glucose agar containing 125 µM tetracycline. Before initiating the two-phase flask cultivation, cells were pre-cultured in MES-buffered YNB-glucose to exponential phase (OD$_{600}$ between 1 to 4) and collected by centrifugation. Collected cells were then resuspended in fresh fermentation medium. To initiate the cultivation, appropriate volumes of pre-cultured cells were transferred to MES-buffered YNB medium with 20 g L$^{-1}$ glucose to an initial OD600 of 0.2 in a total volume of 23 mL medium in a 250 mL flask, and 2 mL sterile dodecane was added after inoculation. Seed culture and test culture were grown in a 200 rpm 30 °C or 37 °C incubator. In the first 12 h of cultivation, 3 ml culture was sampled for growth curve measurement. Dodecane was sampled and stored at −80 °C for terpene analysis.

**Flow cytometry**. Fluorescence in single cells was analysed using a BD Accuri™ C6 flow cytometer (BD Biosciences, USA)[17,39]. For analysis of yEGFP fluorescence, cells sampled from characterisations were directly used for flow cytometry analysis. For analysis of YFAST fluorescence, 100-time-concentrated HMBR was added to the samples to 20 µM final concentration and the sample was mixed before analysis[17]. FSC.H threshold was set at the value of 250,000 for exclusion of debris particles. GFP and/or YFAST fluorescence was excited by a 488 nm laser and monitored through a 530/20 nm bandpass filter (FL1.A), with 10,000 events recorded per sample. Mean values of FSC.A, SSC.A, and FL1.A for all detected events were extracted using a BD Csampler software (BD Accuri C6 software version 1.0.264.21). GFP or YFAST fluorescence level was expressed as the percentage of the average background auto-fluorescence from the exponential-phase cells of GFP-negative reference strain GH4 as described previously[39].

**Metabolite analysis**. The Metabolomics Australia Queensland Node analysed extracellular metabolites. Nerolidol in dodecane samples were analysed as previously described[62]. Dodecane samples (in some cases, diluted with dodecane) were diluted in 40-fold volume of ethanol. The ethanol-diluted samples (20 µL) were injected. A Zorbax Extend C18 column (4.6 × 150 mm, 3.5 µm, Agilent PN: 763953-902) equipped with a guard column (SecurityGuard Gemini C18, Phenomenex PN: AJO-7597) was used. Analytes were eluted at 35 °C at 0.9 mL/min using the mixture of solvent A (water) and solvent B (45% acetonitrile, 45% methanol, and 10% water), with a linear gradient of 5–100% solvent B from 0–24 min, then 100% from 24–30 min, and finally 5% from 30.1–35 min. Analytes of interest were monitored using a diode array detector (Agilent DAD SL, G1315C) at 202 nm wavelength.

**Statistics and reproducibility**. Two-tailed Welch's *t*-test was used for comparing two groups. Data generated from two-to-four biological replicates are presented. Construction-in-processing strains and strain construction procedures are not replicated.

**Reporting summary**. Further information on research design is available in the Nature Research Reporting Summary linked to this article.

## Data availability
Source data are provided in Supplementary Data 1. Any remaining information can be obtained from the corresponding author upon reasonable request.

## Materials availability
Plasmids used in this study are available on Addgene (under submission).

## Code availability
Custom code or mathematical algorithm was not used in this study.

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

## Acknowledgements

BP and this research were supported by a CSIRO Synthetic Biology Future Science Fellowship and the University of Queensland. BP and CEV acknowledge current support from ARC Centre of Excellence in Synthetic Biology and Queensland University of Technology. Metabolite analysis was performed by Dr Manual Plan at Metabolomics Australia Queensland Node. Metabolomics Australia is supported by BioPlatforms Australia through the Commonwealth government's National Collaborative Research Infrastructure Strategy (NCRIS). Yeast strains in this study derive from CEN.PK background strains, which were provided by EUROSCARF (Scientific Research and Development GmbH, Germany) under a non-commercial licence.

## Author contributions

B.P., C.E.V. and G.D. contributed to early-stage conception of the project. C.S., M.T. and C.B.H. provided advisory opinions on the project. C.E.V., G.D. and M.T. supported and coordinated the running of the project. B.P. performed the experiments. N.C.B. and Z.L. participated in strain construction and characterisation. B.P. prepared the draft of the manuscript. C.E.V. revised manuscript. G.D., C.S., M.T. and C.B.H. participated in manuscript revision. All authors contributed to result analysis and discussion.

## Competing interests

The authors declare no competing interests.
