## [Transparent Peer Review File · Communications Biology]

This manuscript has been previously reviewed at another Nature Portfolio journal. This document only contains reviewer comments and rebuttal letters for versions considered at Communications Biology.

REVIEWERS' COMMENTS:

Reviewer #1 (Remarks to the Author):

The authors have prepared a substantially modified manuscript, with a much clearer story, representing a solid piece of work in the field. The work to evaluate and engineer new promoter systems is clear and well presented. With the updated story it is clear that the authors are trying to understand how to get "well behaved" inducible promoter systems.

One point: There is a lot of recent work, largely unreferenced in this manuscript where "well behaved" is called robust, ie robust production or robust expression. The authors should consider relating the current work in this context (the word robust is only used once in the current work) The choice of HAC1, for example, was due to its robustness or environmental insensitivity.

Additionally, there the manuscript should be checked for small errors, and a few places to increase clarity:

1) Figure (amonia ammonia)

2) Figure 2: Better illustration of the modified promoter sequences which include the transactivation domain. The nomenclature is not readily clear in the figure and not explained in the legend. Black squares, green triangles?

Figure 2e is referred to in the text but does not exist.

3) Figure 3, again green triangles are not defined.

Reviewer #2 (Remarks to the Author):

This is an excellent manuscript with impactful findings. Since this is a submission forwarded from another Nature journal, the authors have addressed all previous comments and there is nothing further to add.

Response to Reviewers' comments

Reviewer #1 (Remarks to the Author):

The authors have prepared a substantially modified manuscript, with a much clearer story, representing a solid piece of work in the field. The work to evaluate and engineer new promoter systems is clear and well presented. With the updated story it is clear that the authors are trying to understand how to get "well behaved" inducible promoter systems.

Response: Thanks very much for this encouraging comment!

(1) One point: There is a lot of recent work, largely unreferenced in this manuscript where "well behaved" is called robust, ie robust production or robust expression. The authors should consider relating the current work in this context (the word robust is only used once in the current work) The choice of HAC1, for example, was due to its robustness or environmental insensitivity.

Response: Thanks for pointing out this omission/lack of focus on the emerging technical language/insufficient referencing to recent literature. We have addressed this by adding the following sentence in the introduction at the top of page 4:

Consequently, significant work has been devoted to development of robust, predictable genetic componentry in yeast ^{6, 35, 36}.

Additionally, there the manuscript should be checked for small errors, and a few places to increase clarity:

(2) 1) Figure (amonia ammonia)

Response: Revised as suggested

(3) 2) Figure 2: Better illustration of the modified promoter sequences which include the transactivation domain. The nomenclature is not readily clear in the figure and not explained in the legend. Black squares, green triangles?

Response: We have improved Figure 2 by defining DNA binding domain (DBD) and trans-activating domain (TA) in the figure and legend, removing black squares, and defining green triangles. We hope this is better now!

(4) Figure 2e is referred to in the text but does not exist.

Response: This was a 'hangover' from a previous version. We have removed the incorrect cross-reference in the text.

3) Figure 3, again green triangles are not defined.

Response: There are no green triangles in Figure 3 – perhaps the reviewer meant the green cross? This is labelled 'GAL4'.

Reviewer #2 (Remarks to the Author):

This is an excellent manuscript with impactful findings. Since this is a submission forwarded from another Nature journal, the authors have addressed all previous comments and there is nothing further to add.

Response: Thank you very much for this encouraging comment!